# Capsule-based federated reinforcement learning adaptive sliding mode for anomaly detection and control of floating wind turbines

**Hadi Mohammadian KhalafAnsar**[☯¤], **Jafar Keighobadi**[iD][☯¤*], **Mohsen Shahhosseini**[☯¤]

Faculty of Mechanical Engineering, University of Tabriz, Tabriz, East Azerbaijan, Islamic Republic of Iran

☯ These authors contributed equally to this work.
¤ Current Address: University of Tabriz, Tabriz, 29 Bahman Blvd, Iran
\* keighobadi@tabrizu.ac.ir

## Abstract

Floating wind turbines (FWTs) are now recognized as one of the most effective and affordable renewable energy sources. However, their performance is strongly influenced by dynamic environmental conditions, particularly sea waves under significant oscillatory conditions. Ocean wave and wind disturbance affect turbine positioning, underscoring the critical essential for adaptive and robust control mechanisms to manage the unpredictable external inputs. In this context, we present an innovative method based on federated deep learning for training capsule networks to detect disturbances and enable adaptive robust control of FWTs among the environmental uncertainty. Through the proposed technique, a unique mixture of sliding mode control and deep reinforcement learning (DRL) yields in the extraction of wide features and modeling of spatial relationships between sensor data in the capsule networks framework. Furthermore, by employing federated learning, the capsule-net model is trained in a distributed manner across multiple wind turbines. Therefore, enhanced accuracy and effectiveness of disturbance detection are guaranteed. Simulation results reveal effective identification of disturbances which in turn improves the performance and stability of FWTs under the coarse environmental situation. The global Lyapunov stability analysis proves the FWTs' closed-loop stability. Performance of the superior DRL is evaluated in comparison with a radial basis function neural network (RBFNN) estimation. The innovative DRL method represents a significant advancement in the control of FWTs as a high potential of development for intelligent management of similar systems. As a final aim, this research work finds out the reliability and efficiency of FWTs in variable weather conditions (short-term) and erratic ocean environments (long-term). Moreover, the control system makes a substantial impact on the sustainable development of the wind and renewable energy sector.

**Data availability statement:** All relevant data are within the manuscript. The minimal dataset necessary to replicate the study findings has been deposited in the Zenodo repository (https://zenodo.org/) under DOI: [https://doi.org/10.5281/zenodo.17519490].

**Funding:** The author(s) received no specific funding for this work.

**Competing interests:** The authors have declared that no competing interests exist.

## Introduction

Fossil fuels as the worldwide energy source along recent decades lead to polluted air, soil degradation, groundwater pollution, and greenhouse gas emissions that directly threaten the health of ecosystems and human communities. The combustion of fuels releases about 35 gigatons of carbon dioxide into the atmosphere annually, which accounts for nearly 89 percent of total global emissions. Given the lower level of natural processes, like ocean absorption, in reduction of this destructive phenomenon, the carbon concentration increases in the atmosphere each year. Such a growth in greenhouse gases brings some consequences including global warming, ocean acidification, and significant climate changes. The impact on human life is also significant; air pollution caused by burning fossil fuels affects millions of lives annually, and it is estimated that reduction in the use of these resources could save 3.6 million lives worldwide each year [1–3].

To overcome the mentioned threats, the global community has initiated a significant transition from fossil fuels to renewable energy sources. International agreements, such as the Paris Agreement and the United Nations Sustainable Development Goals, put emphasis on the requirement of decreasing dependence on non-renewable energy and substitute clean energy. Countries are investing in wind, solar, geothermal, and hydroelectric power projects to balance some energy demand currently afforded by fossil fuels. Renewable energy sources now comprise approximately 18 percent of total energy consumption, and this share is likely to increase significantly in the next decades. Among the alternative technologies, FWTs have emerged as a promising innovation, empowering the harnessing of strong and consistent winds in deep-water areas. This reduces reliance on shore-founded installations and minimizes the visual impact of offshore wind farms. Furthermore, relocating wind farms farther offshore not only provides access to more stable wind resources but also enhances opportunities for new transportation routes and fisheries [1–3].

Despite their abundant advantages, the steady process of FWTs encounters momentous challenges owing to the high variability of wind speeds, wave turbulence, and environmental degradation. Therefore, the development of progressive control methods is essential to ensure such systems remain dynamic and energy-efficient. Various methods have been proposed in recent years to enhance the reliability of FWTs. Among them, fractional-order sliding mode controllers (FOSMC) have obtained particular importance for their ability to improve transient response and keep the system stable in the presence of disturbances. Additionally, the application of fractional calculus in load frequency control (LFC) of island microgrids has validated that the above methods can more effectively overturn fluctuations caused by renewable energy sources. The use of meta-heuristic algorithms, such as the Sine Cosine Algorithm (SCA) and Harmony Search (HS), to fine-tune control parameters also enables efficient system control under unstable conditions [4].

Based on the mentioned techniques, further research explored fractional-order control methods for chaotic systems, such as stabilizing laser systems using fractional-order PID sliding mode control (SMC). This approach employs fractional

derivatives to decrease chattering noise, to cause smoother control and to enhance the stability without requiring many computational resources. Moreover, adaptive-SMC was developed to address synchronization challenges in fractional-order chaotic neural network systems, overcoming issues such as input saturation and delays commonly encountered in real-world applications. Recently, these methods have been extended to synchronize chaotic systems in power networks, providing a promising foundation for controlling FWTs, where synchronization and stability are critical [5–7].

More generally, traditional feedback control techniques face problems in systems that are subject to multiple non-linearities and externalities. Consequently, adaptive control systems are used which automatically adjust the controller settings in response to changes in the system's conditions. Sliding-mode controllers (SMC) are usually used when stability and resilience to failure are a priority. These controllers use sliding techniques and include adaptive mechanisms to ensure the stability of the system. Intelligent control algorithms, such as neural networks and fuzzy logic, have been developed for managing the uncertainties in the system's performance and have proved their worth in a wide range of applications. In the area of floating wind turbines, adaptive control algorithms, including adaptive sliding mode control and neural network controllers, have been developed. Recent studies have also examined the use of reinforcement learning (RL) to control variable-speed wind turbines, adapt controllers to fluctuating wind conditions, and manage doubly fed induction generators in real-time. Hybrid intelligent controllers that combine conventional PID regulators with fuzzy systems have shown promising results in reducing turbine loads and increasing power output [8–15]. In addition to control problems, accurate modelling of FWT dynamics is an active field of research. Recent work on fractional time equations and mesh-free methods has provided promising approaches to modelling FWT. Polynomial fractional time equations, particularly those using the Caputo derivative, offer great flexibility for dynamic systems modelling [16]. For complex geometry, mesh-free methods such as the Local Radial Point Interpolation Method (RPIM) are ideal, but may be computationally expensive [17]. Another option is the Predynamic Differential Operator (PDO), which is good for simulating complex, highly interacting systems such as FWTs. However, the defining functions at each node may increase computational complexity [18]. Spectral collocation methods, especially those utilizing polynomials, show potential to solve time-fractional diffusion equations with stochastic time-fractional coefficients, and to model unpredictable and noisy dynamics of FWTs. These methods offer high accuracy; they come with computational costs and sensitivity irregular data [8].

However, a review of the technical literature reveals a clear gap in the research: the lack of a unified control framework combining the high reliability of conventional controllers with the advanced intelligence and distributed learning capabilities of FWTs. To fill the gap and provide a comprehensive solution, this study proposes a multi-faceted approach to the control and stabilization of floating wind turbines. This research develops an adaptive sliding mode controller using deep reinforcement learning and real time data feeding of dynamic tuning. The main innovations of this research are listed as:

1. Use of capsule networks for precise identification of disturbances instead of traditional neural networks

2. Application of federated learning for distributed training of controllers in order to preserve data privacy and increase scalability

3. Integration of sliding mode control with deep reinforcement learning to increase the stability and adaptability of the system to different environmental conditions.

4. These collective innovations cause to decrease of unwanted oscillations as chattering phenomena and increase of durability of mechanical structures, more energy efficiency. Besides, robust stability in the presence of uncertainty and disturbances is acquired based on the Lyapunov method. Hence, the results of this research work may serve as an effective step toward the sustainable development of renewable energy and the global movement to reduce dependence on fossil fuels.

 

## Materials and methods

The main objective of this research is to design a control system that can keep the FWT stable against uncertain and variable conditions of waves and wind in ocean. Our control system is a hybrid framework that leverages the power and stability of classical control methods along with the captured intelligence of machine learning. The components of this framework are explained in a logical order as following steps of an applicable algorithm. Then, we go in details. This research did not require ethics approval as it involved only computational simulations and theoretical analysis without any human or animal participation.

Step 1: Control Foundation - Adaptive Sliding Mode Control (ASMC)

The main foundation of our controller is constructed based on SMC as an ideal choice for under disturbance offshore FWTs owing to robustness against external disturbances and model uncertainties. In simple terms, the SMC push the plant to move on a predefined and stable sliding surface path and stay on the surface as well. In a standard SMC, we need to have in advance the maximum possible disturbance bound, which is an unknown value in practice. To solve this problem, we design an adaptive control mechanism in which the controller should adjust characteristic parameters online. Therefore, according to the changing conditions tolerating the hypothesized offshore FWT, a real time control process is carried out without a priori knowledge of the disturbances.

In the main term $-\rho||s||s$, of the SMC responsible for trajectory tracking in the presence of disturbances and uncertainties, we aim intelligent and optimal estimate of the parameter $\rho$ which represents the disturbance index at each moment.

Step 2: The Brain - Deep Reinforcement Learning (DRL) Neural Network

To solve the challenge of estimating $\rho$, we design a "smart brain" for the SMC gathered with the DRL. This neural network continuously observes the turbine state through sensor data of rotation, yaw angle and accordingly learns how to best adjust the parameter $\rho$ to both maintain stability and prevent unwanted chattering oscillations. This approach transforms our controller from a merely robust system to an intelligent and self-regulating system.

Step 3: Innovations in network architecture

To increase the efficiency of this "smart brain," we implemented two key innovations in its architecture:

1. Using a capsule network for accurate disturbance detection

Instead of using conventional neural networks, we used a capsule network to analyze the input data from sensors. Turbine sensor data for surge, sway, pitch, roll, heave, and yaw have complex spatial relationships with each other. Capsule networks, unlike traditional networks, are able to understand these hierarchical and spatial relationships. This feature allows complex disturbances caused by wind and waves to be detected with much higher accuracy.

2. Using federated learning for distributed training

In a wind farm, several turbines are operating simultaneously. To train a powerful model, data from all turbines would be used. Continuously sending huge amounts of raw data from each turbine to a central server is both costly and privacy-insecure. In this method, each turbine trains its own smart model locally through its own data. Then, merely the updated model parameters without the raw data are sent to a central server to be aggregated to form an improved global model. This global model is then fed back to all turbines. This process allows the final model to benefit from the "collective experience" of all turbines without violating data privacy.

Step 4: Mathematical proof of system stability

After designing of a complex system, a fundamental question about the stability of control system remains. Using the Lyapunov Stability Analysis as a standard and powerful mathematical method in control engineering, the system's

performance is assessed by a pseudo-energy function. The positive definite candidate for Lyapunov function yields in a negative definite time derivative showing that the system is stable. Therefore, all internal signals remain bounded, and the tracking error tends to zero, i.e., the controller will steer the turbine exactly to the desired path over time. This mathematical proof ensures the stability and reliability of our proposed control framework.

**Modeling of the online adaptive intelligent control**

In this section, we derive the governing equations prior to design of the feedback control system. Subsequently, an in-detail proof is provided to examine the convergence of the system's weighting parameters and the asymptotic stability of the tracking error, i.e., the off-track of the actual states of system from corresponding desired values. Now, the assumed model of the multi-input FWT system is expressed as [9,14]:

$$\dot{X}(t) = AX(t) + Bu(t) + Bf_m(t) \tag{1}$$

With respect to Eq. (1), the reference model for the aforementioned adaptive control system is applied to generate tracking trajectories:

$$\dot{X}_m(t) = A_m X_m(t) + B_m r(t) \tag{2}$$

We establish the following assumptions for our analysis:

1. The eigenvalues of user defined matrix $A_m$ are located in the left−half of the complex Laplace plane, and the signal function $r(t)$ is energy bounded.

2. There exist constant matrices, defined as $K_1^* \in R^{n \times m}$ and $K_2^* \in R^{m \times m}$, such that:

$$A + BK_1^{*T} = A_m, BK_2^* = B_m$$

3. The matrix $Q \in \mathbb{R}^{m \times m}$ is positive-definite.

$$M = K_2^* Q = (K_2^* Q)^T = Q^T K_2^{*T} > 0$$

4. Consequently, the ultimate upper bound of the uncertainty and disturbance $\rho$, is the next user-defined known and bounded parameter. This quantitative aspect implies that the system's robustness against uncertainties and disturbances is guaranteed as long as they remain within this established bound:

$$f_m \leq \bar{\rho}$$

Regarding the tracking error as [14]:

$$e(t) = X(t) - X_m(t) \tag{3}$$

Taking the time derivate of Eq. (3) gives dynamics of the tracking error as:

$$\dot{e} = A_m e + (A - A_m)X + Bu + Bf_m - B_m r \tag{4}$$

By defining the integral surface as follows:

$$s(t) = \lambda e - \int_0^t \lambda A_m e \, d\tau \tag{5}$$

the sliding surface kinematics is obtained as:

$$\dot{s} = \lambda(A - A_m)X + \lambda Bu + \lambda Bf_m - \lambda B_m r \tag{6}$$

To figure out the equivalent control input, $\dot{s} = 0$ yields [19]:

$$u_{eq} = -(\lambda B)^{-1}\lambda(A - A_m)X + (\lambda B)^{-1}\lambda B_m r - f_m$$
$$= K_1^{*T}X(t) + K_2^* r(t) - f_m \tag{7}$$

where $(\lambda B)^{-1}\lambda(A_m - A) = K_1^{*T}$ and $(\lambda B)^{-1}\lambda B_m = K_2^*$.

In this context, the proposed control signal is formulated as follows:

$$u(t) = K_1^{*T}X(t) + K_2^* r(t) - \rho\frac{s}{\|s\|} \tag{8}$$

To address the innovation of the proposed sliding surface $s(t) = \lambda e - \int_0^t \lambda A_m e \, d\tau$ and the adaptive controller $u(t) = K_1^T(t)X(t) + K_2(t)r(t) - \frac{\rho s}{\|s\|}$, the sliding surface introduces an integral term that dynamically adjusts the system's trajectory by considering the history of states within the sliding window. This enhances robustness against disturbances and uncertainties. Unlike conventional sliding surfaces that rely solely on the current tracking error, the proposed surface allows for reduced sensitivity of the overall control system to sudden changes, as well as high adaptability over time. The designed adaptive controller incorporates real-time adaptation of the gains $K_1(t)$ and $K_2(t)$ in response to variations in the system dynamics and disturbances. The final term $\frac{\rho s}{\|s\|}$ in the control law ensures that the control input effectively counteracts large deviations, which in turn leads to reduced chattering by modulating the control force based on the magnitude of the sliding surface. This approach provides a more nuanced response compared to conventional adaptive controllers, resulting in improved stability and performance under a broader range of operational conditions.

SMC is well-regarded for its robustness against uncertainties and external disturbances, making it a natural choice for controlling the complex dynamics of FWTs. However, standard SMC can suffer from the chattering phenomenon and requires precise model information, which can be challenging to obtain in real-world scenarios. On the other hand, DRL excels at learning optimal control strategies from data, adapting to changing conditions, and managing complex, high-dimensional systems. By integrating DRL with SMC, we aimed to create a control system that leverages the robustness of SMC while utilizing DRL to adaptively tune the controller, reducing chattering and improving performance under varying operational conditions. This approach addresses the specific challenge of maintaining stability and performance in the highly uncertain and dynamic environment of FWTs, where traditional control methods may fall short. By implementing an adaptive sliding mode (ASM) controller with DRL, the hybrid system becomes robust against any uncertainties or disturbances. Fig 1 depicts the block diagram of the developed closed-loop system in MATLAB© Simulink. The implementation comprises several essential blocks, detailed as follows:

The adaptive laws are used to calculate the coefficients in Eq. (8). This block plays a key role in the dynamic adjustment of the controller parameters according to changing system conditions. The controller block is responsible for computing the final control signal after receiving the coefficients from the Adaptation Block. It is a key component of the closed-loop system and ensures effective control in response to the dynamic nature of uncertainties and disturbances. The hierarchical relationship among the blocks in MATLAB© Simulink offers profound insight into the holistic approach used to address uncertainties and manage system inconsistencies. The described methodology combines the advantages

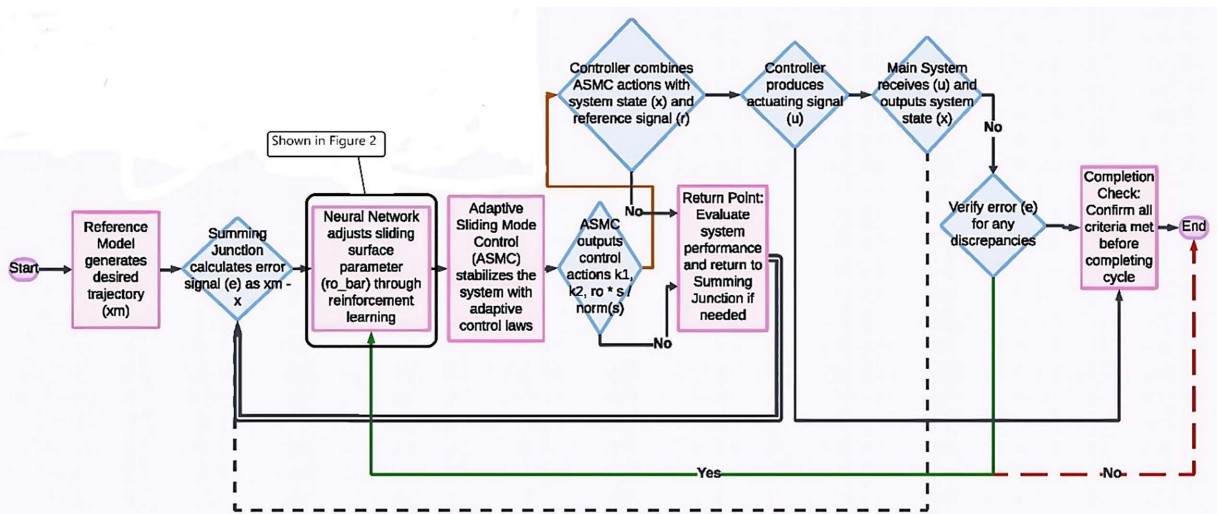

**Fig 1. The block diagram of the simulated system in MATLAB©.**

of adaptive control in sliding mode and DRL, resulting in a control system that dynamically responds to the complexities of real-world applications. The DRL neural network block estimates the upper bounds of uncertainty and interference based on adaptive laws derived from the Lyapunov function.

The Lyapunov function plays a central role in deriving the adaptation rules within a system. Therefore, by assuming a Lyapunov candidate function with positive-definite properties, the stability proof is terminated when the derivative of this function becomes negative semi-definite. This is a critical point in the analytical evaluation, where the Lyapunov function serves as the key mathematical tool for determining the stability and convergence characteristics of the adaptive system [9,14].

Substitution of the estimated values $K_1^{*T}$ and $K_2^*$ in nominal control Eq. (8):

$$u(t) = K_1^T(t)X(t) + K_2(t)r(t) - \rho\frac{s}{\|s\|} \tag{9}$$

Substituting this control input into the original system dynamics yields the expected change in the system's behavior.

$$\dot{X}(t) = AX(t) + B\left[K_1^T(t)X(t) + K_2(t)r(t)\right]$$
$$+Bf_m - B\rho\frac{s}{\|s\|}$$
$$= A_mX(t) + B_mr(t)$$
$$+B_m\left[K_2^{*-1}\widetilde{K}_1^T(t)X(t) + K_2^{*-1}\widetilde{K}_2(t)r(t)\right]$$
$$+Bf_m - B\rho\frac{s}{\|s\|} \tag{10}$$

Given the expression above for the derivative of the state variables, we can deduce the dynamics of the tracking error:

$$\dot{e}(t) = A_me(t) + B_m\left[K_2^{*-1}\widetilde{K}_1^T(t)X(t) + K_2^{*-1}\widetilde{K}_2(t)r(t)\right]$$

$$+Bf_m - B\rho\frac{s}{\|s\|} \tag{11}$$

## DRL neural network

DRL value will be utilized to estimate the uncertainty and disturbance upper limit denoted by $\bar{\rho}$. The neural network employed for this purpose comprises several layers, as illustrated in Fig 2. This study employs a hybrid architecture combining a Capsule Network and DRL to estimate the disturbance effects on a floating wind turbine. The model is trained using federated learning and leverages the advantages of the Capsule Network for hierarchical modeling and robustness to perturbations. The input consists of six system status variables collected from the wind turbine's sensors. These variables include surge, sway, heave, roll, pitch and yaw. The input data is represented by the vector:

$$\boldsymbol{X} = [x_1, x_2, \ldots, x_6]$$

where $\boldsymbol{X}$ is the vector of the sensor data and index 6 is the number of input elements. The orange capsules are processing the input functions as shown. Each capsule stores data on part of the spatial structure as a vector. The properties of each layer $\boldsymbol{u}_i$ are mapped to the predictions of the next layer by means of the transformation matrix $\boldsymbol{V}_{j|i} = \boldsymbol{W}_{ij} \cdot \boldsymbol{u}_i$. In a capsule network, the routing-by-assignment mechanism adjusts the routing weight $r_{ij}$ according to the similarity and the concurrency of the capsules. In this mechanism, the output of each capsule $i$ is transformed into a capsule $j$ by means of a transformation matrix $\boldsymbol{W}_{ij}$. The weighting $r_{ij}$ is adjusted by a concave algorithm between the output vector of the capsules [20].

In the Capsule Network routing process, the initial weights of $r_{ij}$ are usually initialized in a uniform way. These weights are then iteratively adjusted by agreement between the vector(s) of the predictor $\boldsymbol{V}_{j|i}$. The following update rule applies [20]:

$$r_{ij} \propto exp\left(\boldsymbol{u}_i^\top \cdot \boldsymbol{V}_{j|i}\right)$$

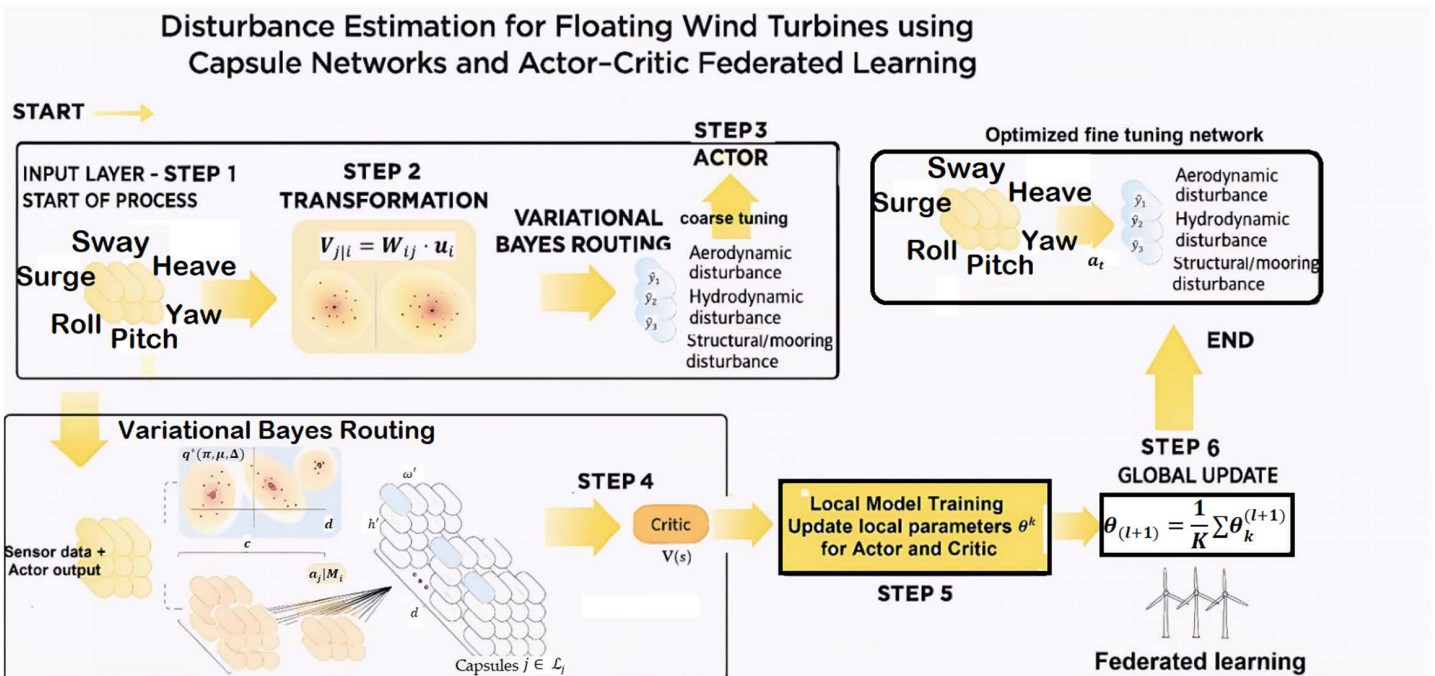

**Fig 2. Disturbance estimation for floating wind turbine using capsule networks and Actor-Critic federated learning.**

The arrows in Fig 2 represent the collection of votes and the transfer of these votes to the output capsules. The agreement between capsule inputs is calculated and the weightings updated. This process continues until convergence or predetermined number of iterations have been achieved.

The middle capsule layer is responsible for modelling the middle level functions. This layer receives information from the convolution layer and transfers it to the output capsule layer. As the arrows in Fig 2 show, the characteristics of the input capsules are converted into predictions of the output capsules (the votes). Transformation matrices are used to perform these transformations $\boldsymbol{M}_i$. Each input capsule votes for a different output capsule. These votes, $\boldsymbol{V}_{j|i}$, are sent to the intermediate layers for combining and fine-tuning. The output of this layer is composed of three capsules which predict the primary error values of the following variables $\hat{y}_1, \hat{y}_2, \hat{y}_3$. Final output is defined by vector:

$$\hat{\boldsymbol{Y}} = [\hat{y}_1, \hat{y}_2, \hat{y}_3]$$

The activation of each output capsule $j$ is calculated as follows:

$$a_j = \sigma\left(\|\boldsymbol{v}_j\|\right)$$

where $\|\boldsymbol{v}_j\|$ is the output vector and σ is the squashing function (similar to the sigmoid function).

The Capsule network acts as an actor in the DRL architecture. The output values of the capsules are called actions. They are defined as follows:

$$\boldsymbol{a}_t = \hat{\boldsymbol{Y}} = f_{\text{Actor}}(\boldsymbol{X}_t)$$

Here, $\boldsymbol{a}_t$ represents the action in step $t$, and $f_{\text{Actor}}$ represents the function of the actor network. The critic network evaluates the performance of the actor and calculates the value function $V(s_t)$ as follows:

$$V(s_t) = f_{\text{Critic}}(\boldsymbol{X}_t, \boldsymbol{a}_t)$$

where $f_{\text{Critic}}$ is the function of the critical network.

The Federated learning is integrated in the network modelling as shown in Fig 2 and a pseudo code for this algorithm is given in Table 1 where $w_t$ are the weights of the model in round $t$, $n_k$ is the number of data samples in client $k$, $\mathcal{L}(w; b)$ is the cost function of the model for the mini-set $b$. In this algorithm, the server periodically receives the updated models from the clients and calculates their average to create a new model. Each wind turbine trains its local capsule model on the following local data:

$$\theta_k^{(t+1)} = \theta_k^{(t)} - \eta \nabla \mathcal{L}(\theta_k^{(t)}; \boldsymbol{X}_k, \boldsymbol{Y}_k)$$

where $\mathcal{L}$ is cost function and $\theta_k$ stands for turbine model parameters.

Local parameters of the turbines are sent to the central server and summarized:

$$\theta^{(t+1)} = \frac{1}{K} \sum_{k=1}^{K} \theta_k^{(t+1)}$$

where $k$ is the number of turbines.

This hybrid architecture, which makes use of capsule networks and deep reinforcement learning, provides an optimal method for predicting the faults in floating wind turbines.

**Table 1. Federated averaging algorithm pseudocode.**

In this algorithm, we have $K$ customers, which are specified by index $k$; $B$ is the local minibatch size, $E$ is the number of local epochs, and $\eta$ is the learning rate.

| Server operation: |
|---|
| 1. Determine the initial value $\mathbf{w_0}$. |
| 2. For each round ($\mathbf{t} = \mathbf{1}, \mathbf{2}, \ldots$) do: |
| - Choose $\mathbf{m}$ randomly: $\mathbf{m} \leftarrow \mathbf{max}(\mathbf{C \cdot K}, \mathbf{1})$ |
| - Choose the set $\mathbf{S_t}$ randomly from $m$ customers. |
| 3. For each customer $\mathbf{k}$ who is a member of the set $\mathbf{S_t}$ do in parallel: |
| - $\mathbf{w_{t+1}^k} \leftarrow \mathbf{ClientUpdate}(\mathbf{k}, \mathbf{w_t})$ |
| 4. Averaging customer updates: |
| - $\mathbf{m_t} \leftarrow \sum_{\mathbf{k} \in \mathbf{S_t}} \mathbf{n_k}$ |
| - $\mathbf{w_{t+1}} \leftarrow \sum_{\mathbf{k} \in \mathbf{S_t}} \frac{\mathbf{n_k}}{\mathbf{m_t}} \cdot \mathbf{w_{t+1}^k}$ |
| Client update operation ($\mathbf{ClientUpdate}(\mathbf{k}, \mathbf{w})$): |
| 1. Divide the data set $\mathbf{P_k}$ into mini-sets of size $\mathbf{B}$. |
| 2. For each local epoch $\mathbf{i}$ from 1 to $E$ do: |
| - For each mini-set $\mathbf{b}$ of the set $\mathbf{B}$ |
| - $\mathbf{w} \leftarrow \mathbf{w} - \eta \cdot \nabla \mathbf{L}(\mathbf{w}; \mathbf{b})$ |
| 3. After the epochs end, return the final weight $\mathbf{w}$ to the server. |

## Stability analysis

The final threshold for interference is expected to be:

$$\hat{\bar{\rho}}(\mathbf{x}, \omega) = \hat{\omega}^T \phi(\mathbf{x}) \tag{12}$$

In the described system, the input is the state variables, and the coefficients are given by $\hat{\omega}^T$ and serve as weights. The output of the second layer is equal to the function $\phi(\mathbf{x})$.

Several key assumptions are taken into account according to [9,14,19], which is prevalent in literature review:

1. Optimal network weight values shall be determined by meeting the following equality criteria:

$$\omega^{*T}\phi(\mathbf{x}) - \bar{\rho}(t) = \varepsilon(\mathbf{x}) \ \ \forall \ |\varepsilon(\mathbf{x})| < \varepsilon_1 \tag{13}$$

These assumptions are an additional basis for analyzing and investigating the behavior of the system and the optimization criteria.

2. The final threshold value relates to the following inequality:

$$\bar{\rho}(t) - f_m > \varepsilon_0 > \varepsilon_1 \tag{14}$$

Based on the derivative of the sliding surface as expressed in Eq. (15), we can determine the Lyapunov function as shown in Eq. (16):

$$\dot{s}(t) = \lambda B_m \left[ K_2^{*-1} \widetilde{K}_1^T(t) X(t) + K_2^{*-1} \widetilde{K}_2(t) r(t) \right] + \lambda B f_m - \lambda B \rho \frac{s}{\|s\|} \tag{15}$$

$$V = \tfrac{1}{2}s^T s + \tfrac{1}{2}tr\left[\widetilde{K}_1 M^{-1}\widetilde{K}_1^T\right]$$
$$+\tfrac{1}{2}tr\left[\widetilde{K}_2 M^{-1}\widetilde{K}_2^T\right] + \tfrac{1}{2}\eta^{-1}\mu\widetilde{\omega}^T\widetilde{\omega} \tag{16}$$

This formulation is motivated by the desire to use the knowledge obtained from the sliding surface derivative to develop a Lyapunov function that is consistent with the stability analysis of the system. This encourages careful consideration of system dynamics and allows for evaluation of stability parameters. The choice to base the Lyapunov function on the derivative of the sliding surface shows a structured and systematic analysis which makes the theoretical framework robust and reliable.

We consider the selection process as an additional criterion and highlight its positive definitional nature. In this context, we envisage the following:

$$\widetilde{\omega} = \omega^* - \hat{\omega}, \eta = \varepsilon_0 - \varepsilon_1 > 0, M = M^T > 0$$

Taking the derivative of the expression Eq. (16) in time yields:

$$\dot{V} = s^T\dot{s} + tr\left[\widetilde{K}_1 M^{-1}\dot{\widetilde{K}}_1^T\right] + tr\left[\widetilde{K}_2 M^{-1}\dot{\widetilde{K}}_2^T\right]$$
$$-\eta^{-1}\sigma\widetilde{\omega}^T\dot{\hat{\omega}}$$
$$= s^T\left\{\lambda B_m\left[K_2^{*-1}\widetilde{K}_1^T(t)X(t) + K_2^{*-1}\widetilde{K}_2(t)r(t)\right]\right.$$
$$\left.+\lambda Bf_m - \lambda B\rho\tfrac{s}{\|s\|}s\right\} + tr\left[\widetilde{K}_1 M^{-1}\dot{\widetilde{K}}_1^T\right] + tr\left[\widetilde{K}_2 M^{-1}\dot{\widetilde{K}}_2^T\right]$$
$$-\eta^{-1}\sigma\widetilde{\omega}^T\dot{\hat{\omega}} \tag{17}$$

Substituting Eq. (15) in Eq. (17) results in:

$$\dot{V} = s^T\left(\lambda Bf_m - \lambda B\rho\frac{s}{\|s\|}\right) + \lambda B_m\left[K_2^{*-1}\widetilde{K}_1^T(t)X(t) + K_2^{*-1}\widetilde{K}_2(t)r(t)\right]$$

$$+tr\left[\widetilde{K}_1 M^{-1}\dot{K}_1^T\right] + tr\left[\widetilde{K}_2 M^{-1}\dot{K}_2^T\right] - \eta^{-1}\mu\widetilde{\omega}^T\hat{\omega}$$

$$= s^T\left(\lambda Bf_m - \lambda B\rho\frac{s}{\|s\|}\right) + s^T\left\{\lambda B_m K_2^{*-1}\widetilde{K}_1^T(t)X(t) + tr\left[\widetilde{K}_1 M^{-1}\dot{K}_1^T\right]\right\}$$

$$+s^T\left\{\lambda B_m K_2^{*-1}\widetilde{K}_2(t)r(t) + tr\left[\widetilde{K}_2 M^{-1}\dot{K}_2^T\right]\right\} \tag{18}$$

For simplicity, the terms in brackets are set to zero. The traces of the internal expressions are expected to be equal, since the expressions concerned must maintain an internal trace balance. The reflection ($P = \lambda B_m K_2^{*-1}\widetilde{K}_1^T(t)$) returns the following:

$$s^T PX(t) = tr\left(PXs^T\right) \tag{19}$$

The following results are obtained from Eq. (19) to calculate the values of $\widetilde{K}_1$ and $\widetilde{K}_2$ using the invariance property of the constant, i.e., applying a certain operation to the constant does not change its value.

$$\dot{\tilde{K}}_1^T(t) = \dot{K}_1^T(t) = -Q^T B_m^T \lambda^T s X^T$$

$$\dot{\tilde{K}}_2(t) = \dot{K}_2(t) = -Q^T B_m^T \lambda^T s r^T \tag{20}$$

Thus, the following results are obtained from the simplification of the adaptive rules Eq. (20). Substituting the rules into the derivative of the Lyapunov function, the resulting expression will be:

$$\dot{V} = s^T \left( \lambda B f_m - \lambda B \rho \frac{s}{\|s\|} \right) - \eta^{-1} \mu \tilde{\omega}^T \dot{\hat{\omega}} =$$
$$s^T \left( \lambda B f_m + \lambda B \bar{\rho} - \lambda B \bar{\rho} - \lambda B \rho \frac{s}{\|s\|} \right) - \eta^{-1} \mu \tilde{\omega}^T \dot{\hat{\omega}} \tag{21}$$

Given the above data in which for each variable it is set that it is less or equal to the norm value of the corresponding variable, it can be concluded that the next result will look like:

$$\dot{V} \leq \| s \| \left( \| \lambda B \| f_m + \| \lambda B \| \bar{\rho} - \| \lambda B \| \bar{\rho} - \| \lambda B \| \rho \frac{s}{\| s \|} \right)$$

$$-\eta^{-1} \mu \tilde{\omega}^T \dot{\hat{\omega}} = \| s \| \| \lambda B \| (f_m + \bar{\rho} - \bar{\rho}) - \| \lambda B \| \rho \| s \|$$

$$-\eta^{-1} \mu \tilde{\omega}^T \dot{\hat{\omega}} = - \| s \| \| \lambda B \| (\bar{\rho} - f_m) + \| \lambda B \| \| s \| (\bar{\rho} - \rho) -$$

$$\eta^{-1} \mu \tilde{\omega}^T \dot{\hat{\omega}} = - \| s \| \| \lambda B \| (\bar{\rho} - f_m) + \| \lambda B \| \| s \| \left( \omega^{*T} \phi - \varepsilon - \hat{\omega}^T \phi \right)$$

$$-\eta^{-1} \mu \tilde{\omega}^T \dot{\hat{\omega}} = - \| s \| \| \lambda B \| (\bar{\rho} - f_m) - \| \lambda B \| \| s \| \varepsilon$$

$$+ \left[ \| \lambda B \| \| s \| \tilde{\omega} \phi - \eta^{-1} \mu \tilde{\omega}^T \dot{\hat{\omega}} \right] \tag{22}$$

Where $\bar{\rho}$ and $\rho$ were replaced by $\omega^{*T}\phi - \varepsilon$ and $\hat{\omega}^T \phi$, correspondingly using Eqs. (12) and (13). The value inside the brackets in this specific case is set equal to zero to obtain the formula allowing to express the adaptive rule for changing the weights in the neural network:

$$\dot{\hat{\omega}} = \eta \| s \| \phi(x) \tag{23}$$

The reasoning is continued with excluding the weight factor from the derivative of the Lyapunov function, combined with the norm inequality surpassing the value of the variable. The purpose of this method is to differentiate and improve the analytical logic, while creating an opportunity to include more sophisticated analysis of mathematical issues:

$$\begin{aligned}
\dot{V} &= -\parallel s \parallel \parallel \lambda B \parallel (\bar{\rho} - f_m) - \parallel \lambda B \parallel \parallel s \parallel \varepsilon \\
&\leq -\parallel s \parallel \parallel \lambda B \parallel (\bar{\rho} - f_m) + \parallel \lambda B \parallel \parallel s \parallel \varepsilon \\
&= -\parallel s \parallel \parallel \lambda B \parallel \varepsilon_0 + \parallel \lambda B \parallel \parallel s \parallel \varepsilon \\
&\leq -\parallel s \parallel \parallel \lambda B \parallel \varepsilon_0 + \parallel \lambda B \parallel \parallel s \parallel |\varepsilon| \\
&\leq -\parallel s \parallel \parallel \lambda B \parallel \varepsilon_0 + \parallel \lambda B \parallel \parallel s \parallel \varepsilon_1 \\
&= -\parallel s \parallel \parallel \lambda B \parallel (\varepsilon_0 - \varepsilon_1) = -\eta \parallel s \parallel \parallel \lambda B \parallel \leq 0
\end{aligned}$$

(24)

Given the negative sign of the derivative of the Lyapunov function, it is seen that the parameters $s$, $\widetilde{K}_1$ and $\widetilde{K}_2$ converge to zero. Consequently, the values of the parameters are bounded. This deduction is extracted from the derivative equation of the sliding surface that requires bounded. Integrating both sides of Eq. (24) given that:

$$\int_0^t \parallel s \parallel dt \leq \frac{1}{\eta}[V(0) - V(t)]$$

Considering that $V(0)$ and $V(t)$ are finite, the integral is finite as well. Therefore, the combination of the finiteness of the integral and the derivative of the sliding surface, it is possible to conclude that the sliding surface, which is $s$, "will vanish asymptotically" due to Barbalat's lemma. This demonstrates that with the convergence of the sliding surface to zero, the error will tend to zero, which is shown in Eq. (5).

## Results and discussion

The analysis of the FWT is well established, based on the NREL model of design in the USA. Three buoyant cylinders with a triangular shape and a central cylinder for the control tower were designed as a model of a turbine. The whole system weighs 13.5 kilotons and is purposely located 13.46 meters below sea level. Fig 3 represent design highlights of the complexity of the system. The design incorporates aerodynamic forces, buoyancy forces, linear rope forces and drag and inertia forces, each of which generates a torque, but the torque is omitted for reasons of schematic simplicity and focus [21].

In order to formulate the motion equations for the system in question, it is necessary to define the key components, including the state variables named as $x$ and the control inputs named as $u$ and the disturbances introduced by $v$ and $w$ The non-linear function enclosing the system dynamics, named as $f$ can be briefly expressed as [3]:

$$f_Q(x, u, v, w) = \begin{bmatrix} \dot{\vec{x}}_g \\ \dot{\vec{\theta}} \\ \vec{f}_F(x, u, v, w) \\ \vec{f}_T(x, u, v, w) \\ \omega_r \\ f_{1Q}(x, u, v) \\ \omega_g \\ f_{2Q}(x, u, v) \end{bmatrix}$$

(25)

Summing the force equation gives acceleration. In Eq. (25), the sum of all the forces on the structure can be expressed as.

$$\vec{f}_F(x, u, v, w) = (m_g I_{3\times3} + diag[\vec{m}_a])^{-1} \sum_j \vec{F}_j(x, u, v, w)$$

(26)

In the given equation, the symbol $m_g$ represents the total mass of the plant, and the symbol $I_{3\times3}$ represents the unity matrix. The expression for the torque, $f_F(x, u, v, w)$, is calculated as the sum of all the forces acting on the system.

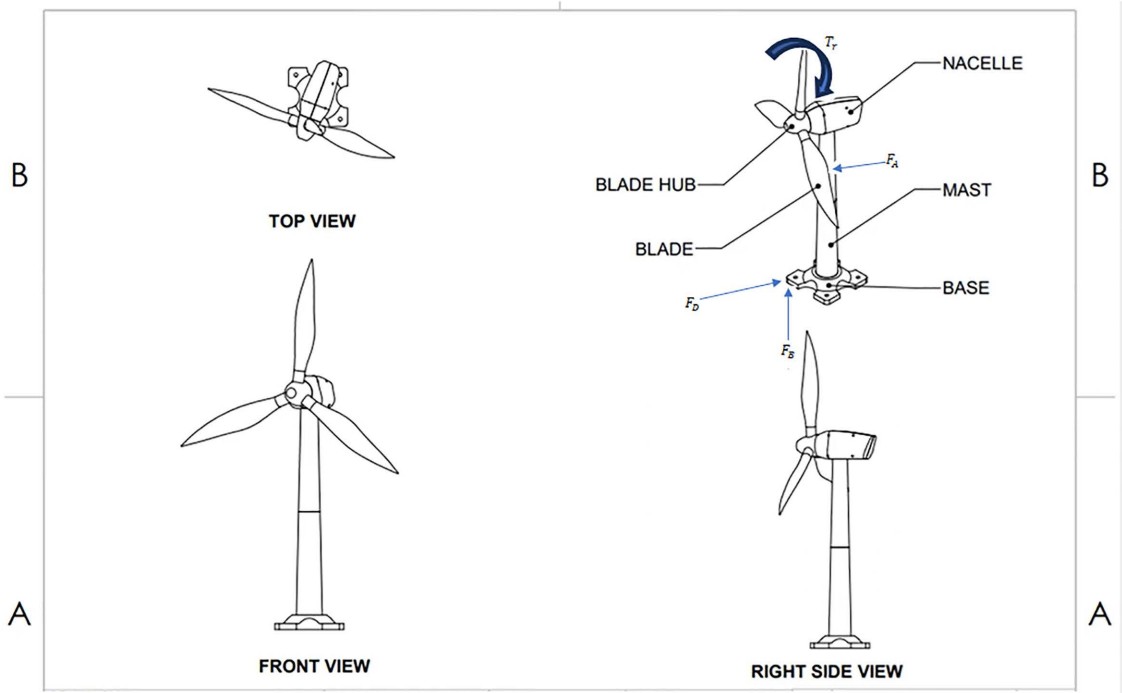

**Fig 3. General diagram of the nonlinear FWT.**

The torque expression, $f_T (x, u, v, w)$, derived from angular accelerations, is computed as the sum of all torques induced by the forces exerted on the plant:

$$\vec{f}_T (x, u, v, w) = \left( R\underline{I}_g^{-1} R^T \right) \sum_j \vec{T}_j (x, u, v, w)$$

(27)

In this context, the symbol $I_g$ represents the tensor of inertia around the vertical axis, while the symbol $R$ represents the matrix of transformation. The term $T_j (x, u, v, w)$ encapsulates all the moments of force that are exerted on a structure. The following derivations are given for the expressions $f_{1Q} (x, u, v)$ and $f_{2Q} (x, u, v,)$.

$$f_{1Q}(x, u, v) = \dot{\vec{\omega}}_r$$
$$f_{2Q}(x, u, v) = \dot{\vec{\omega}}_g$$

(28)

The calculation of aerodynamic efficiency is approached by taking $C_p$ as the coefficient of performance as defined below:

$$P = \tfrac{1}{2}\rho A_r C_p (\lambda, \beta) \vec{v}_n^3$$

(29)

The paper proposes that the defined paths for the required system movements are structured in a way that optimizes the behavior of the system to achieve maximum uniform energy. In mathematical terms, the pursuit of a maximum is conventionally expressed as the determination of the derivative of a function as being equal to zero. Therefore, this principle applies to the derivative of Eq. (29) both for countries and input variables, which results in the following expressions:

$$\delta P(t) = \frac{\partial P(t)}{\partial x}\delta x + \frac{\partial P(t)}{\partial u}\delta u + \frac{\partial P(t)}{\partial v}\delta v = 0 \tag{30}$$

Given the minimal effect of state variable variations on captured energy, it is acceptable to ignore the state component of Eq. (30) and replace the relevant input(s).

$$\frac{\partial P(t)}{\partial \beta}\delta\beta + \frac{\partial P(t)}{\partial \gamma}\delta\gamma + \frac{\partial P(t)}{\partial v}\delta v = 0 \tag{31}$$

The optimization target refers to the variable amount of angle β as the central control target:

$$\delta\beta = -\left(\frac{\partial P(t)}{\partial\gamma}\delta\gamma + \frac{\partial P(t)}{\partial v}\delta v\right) \Big/ \left(\frac{\partial P(t)}{\partial\beta}\right) \tag{32}$$

The adjustment of the generator torque, the final control input, is performed in response to the generator speed fluctuations. In practical applications, the main principle of generator torque regulation is to maintain a constant rotor speed. This operational strategy is in line with the wider objective of achieving stability and control over the dynamic behavior of the system:

$$\dot{\omega}_r = \frac{1}{J_r}\left(\frac{P}{\omega_r} - (N_{GR})T_g\right) = 0 \tag{33}$$

which gives:

$$\delta T_g = \frac{1}{N_{GR}\omega_r}\left(\frac{\partial P(t)}{\partial\beta}\delta\beta + \frac{\partial P(t)}{\partial\gamma}\delta\gamma + \frac{\partial P(t)}{\partial v}\delta v\right) \tag{34}$$

By solving Eq. (34), the computation of the generator torque is achieved. In addition, the ultimate goal was to find an angle of yaw that corresponded to the direction of the wind. However, due to the limitations of the actuator, the use of the average path is required to ensure a smooth and easy implementation of the action. In addition to the nonlinear model controller, the NREL 5-MW controller defines the PI controller for floating offshore wind turbine (FOWT) as follows:

$$k_P(v) = \frac{2J\omega_{r,rat}\zeta_{des}\,\omega_{des}}{N_g\frac{\partial P}{\partial\beta}(v)}$$

and

$$k_I(v) = \frac{J\omega_{r,rat}\omega^2_{des}}{N_g\frac{\partial P}{\partial\beta}(v)}$$

In this equation, $J$ is the inertia of the rotor, $\omega_{r,\,rat}$ is the nominal speed of the rotor, and $N_g$ is the gear ratio. The parameters of the variables $\zeta_{des}$ and $\omega_{des}$ are user-defined tuning variables. In addition, the coefficient of aerodynamic power, $\partial P/\partial\beta(v)$, is a wind speed dependent metric that determines the sensitivity of aerodynamic power to the angle of the blade. Traditionally, this sensitivity component has been assessed by using numerical linearized wind turbine models and an aerodynamic solver such as OpenFAST. Even for those familiar with this approach, obtaining these models and carrying out the subsequent analysis necessary to define this concept may prove to be time-consuming, inter alia. Given the system characteristics in Table 2, figure. Fig 4 shows a MATLAB simulation of FOWT, comparing the nonlinear controller proposed in this work with the PI gain controller and the output oscillation by using the controllers described above. The modest amount of variation between NREL's reported work and our model demonstrates the correctness of the strategy given in this study.

# Table 2. FWT's properties.

| Property | Sign | Value | Unit |
|---|---|---|---|
| Water density | $\rho$ | 1025 | kg/m³ |
| Physical mass | $m_g$ | 14,072,718 | kg |
| Inertia around x-axis | $I_{xx}$ | 1.695e10 | kg.m² |
| Inertia around y-axis | $I_{yy}$ | 1.695e10 | kg.m² |
| Inertia around z-axis | $I_{zz}$ | 1.845e10 | kg.m² |
| Air density | $\rho_a$ | 1.225 | kg/m³ |
| Effective rotor radius | $R_r$ | 62.94 | m |
| Distance vector from FOWT's center to thrust center | $\vec{r}_{gt}$ | $\begin{bmatrix} -5 \\ 0 \\ 99.889 \end{bmatrix}$ | m |
| Rotor Inertia | $J_r$ | 3.5444e7 | kg.m² |
| Generator Inertia | $J_g$ | 5.34116e2 | kg.m² |
| Driveshaft stiffness on rotor side | $k_r$ | 8.676e8 | N.m/rad |
| Driveshaft damping on rotor side | $b_r$ | 6.215e6 | N.m.s/rad |
| Gear ratio | $N_{gr}$ | 97 | – |

In this section we aim to examine the performance of the proposed controller by a thorough analysis of the proposed FWT. In addition, we rewrite the control equations in section Methods and modeling as MATLAB software-adapted, so that further implementation and analysis can be done in Simulink. The process described will allow a thorough and systematic assessment of the advantages and disadvantages of the controller, its behavior under various conditions and the actual dynamic processes taking place in the FWT. The use of MATLAB© and Simulink provides a powerful computational framework in which to perform accurate and insightful simulations to determine the effectiveness of the proposed controller in the timely regulation of the dynamics of FWT.

The dynamic system of FWT of this paper is defined as follows:

$$\dot{X} = AX + Bu$$

where $A$ and $B$ are defined as follows.

$$A = \begin{bmatrix}
0 & 0 & 0 & 0 & 0 & 0 & 1 & 0 & 0 & 0 & 0 & 0 & 0 & 0 & 0 & 0 \\
0 & 0 & 0 & 0 & 0 & 0 & 0 & 1 & 0 & 0 & 0 & 0 & 0 & 0 & 0 & 0 \\
0 & 0 & 0 & 0 & 0 & 0 & 0 & 0 & 1 & 0 & 0 & 0 & 0 & 0 & 0 & 0 \\
0 & 0 & 0 & 0 & 0 & 0 & 0 & 0 & 0 & 1 & 0 & 0 & 0 & 0 & 0 & 0 \\
0 & 0 & 0 & 0 & 0 & 0 & 0 & 0 & 0 & 0 & 1 & 0 & 0 & 0 & 0 & 0 \\
0 & 0 & 0 & 0 & 0 & 0 & 0 & 0 & 0 & 0 & 0 & 1 & 0 & 0 & 0 & 0 \\
-0.00354 & 0.00015 & 0.0082 & -0.10061 & 0.21378 & -0.1846 & -0.0356 & -0.00665 & -0.00199 & 0.122141 & -020316 & 0.33399 & 0 & -0.02676 & 0 & 0 \\
0.00015 & -0.00267 & 0.0030 & -0.1578 & -0.03201 & 0.08429 & -0.0067 & -0.02007 & -0.0006 & 0.01348 & -0.0899 & 0.06358 & 0 & -0.01015 & 0 & 0 \\
-0.00017 & -6.07\times10^{-5} & -0.1379 & -0.0462 & 0.1443 & 0.00041 & -0.00162 & -0.00046 & -0.0965 & 0.5567 & 0.4175 & -0.02721 & 0 & 0.002721 & 0 & 0 \\
-1.65\times10^{-6} & 3.07\times10^{-5} & -4.73\times10^{-5} & -0.0635 & -0.0024 & -0.00342 & 0.00017 & 1.64\times10^{-5} & 0.00096 & -0.0791 & -0.00012 & 0.001902 & 0 & 0.00137 & 0 & 0 \\
-4.19\times10^{-5} & 1.54\times10^{-6} & 0.00015 & 0.0054 & -0.0625 & -0.00057 & -0.00028 & -0.00012 & 0.00065 & 0.0004 & -0.09493 & 1.52\times10^{-5} & 0 & -0.00367 & 0 & 0 \\
2.20\times10^{-8} & 5.36\times10^{-7} & 0.00017 & -0.0053 & -0.00467 & -0.0036 & 0.00042 & 8.02\times10^{-5} & -4.49\times10^{-5} & 0.00217 & 0.000243 & -0.02455 & 0 & -6.63\times10^{-6} & 0 & 0 \\
0 & 0 & 0 & 0 & 0 & 0 & 0 & 0 & 0 & 0 & 0 & 0 & 0 & 1 & 0 & 0 \\
0 & 0 & 0 & 0.0491 & -0.1252 & 0.0223 & -0.03338 & -0.01249 & 0.004167 & 1.236427 & -3.30399 & -0.00023 & -24.4913 & -0.55305 & 0.25248 & 0.0018 \\
0 & 0 & 0 & 0 & 0 & 0 & 0 & 0 & 0 & 0 & 0 & 0 & 0 & 0 & 0 & 1 \\
0 & 0 & 0 & 0 & 0 & 0 & 0 & 0 & 0 & 0 & 0 & 0 & 16746.76 & 119.9593 & -172647 & -1.236
\end{bmatrix}$$

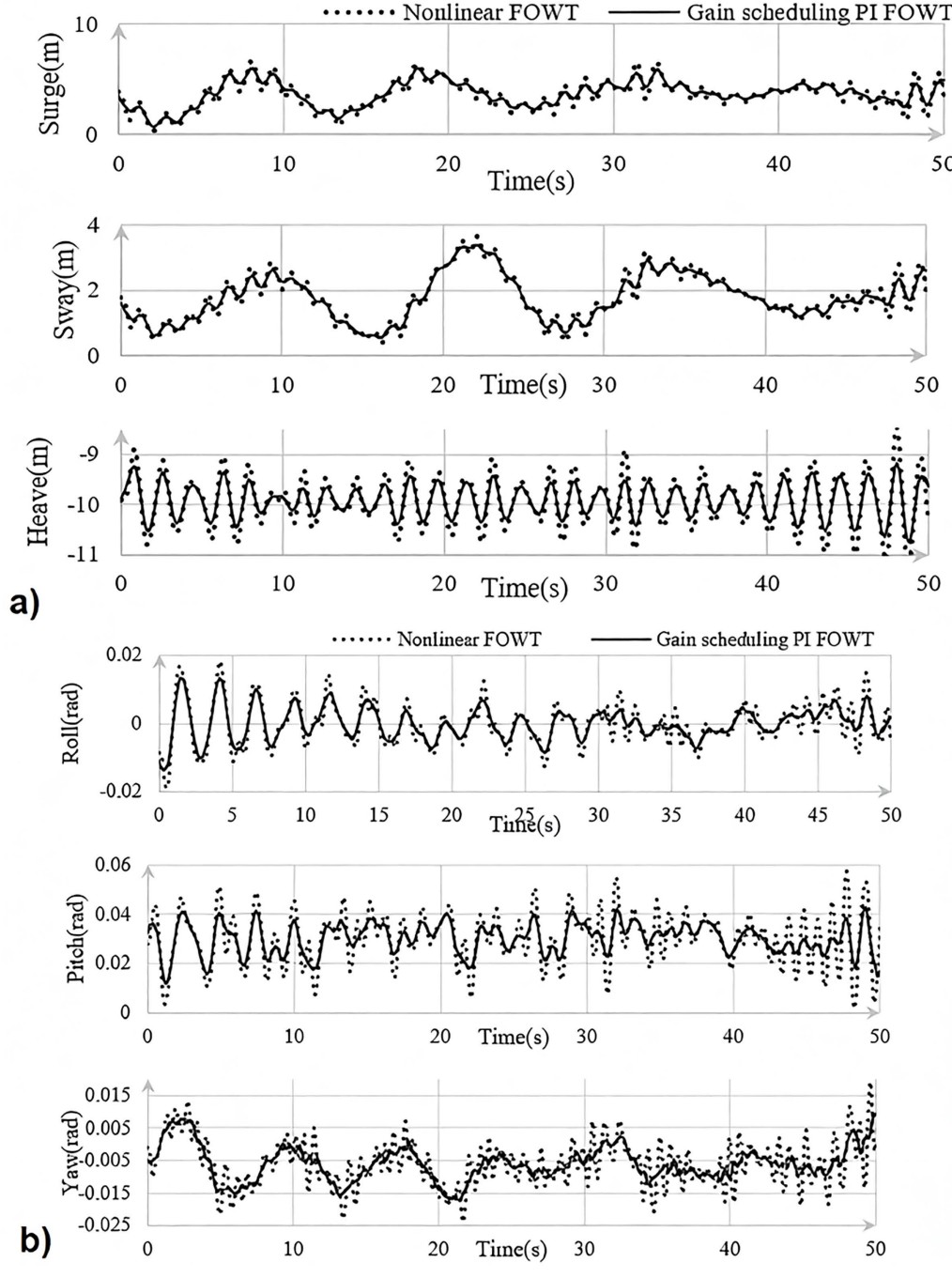

**Fig 4. Desired system simulation in MATLAB for a) translational, b) rotational states.**

$$B = \begin{bmatrix} 0 & 0 & 0 \\ 0 & 0 & 0 \\ 0 & 0 & 0 \\ 0 & 0 & 0 \\ 0 & 0 & 0 \\ 0 & 0 & 0 \\ -0.00293 & 0 & -5.10 \times 10^{-5} \\ -0.00111 & 0 & 0.000242 \\ 0.000298 & 0 & -3.09 \times 10^{-6} \\ 0.00015 & 1.40 \times 10^{-8} & -3.64 \times 10^{-5} \\ -0.0004 & 5.25 \times 10^{-9} & 3.00 \times 10^{-6} \\ -7.27 \times 10^{-7} & -1.64 \times 10^{-9} & 9.89 \times 10^{-7} \\ 0 & 0 & 0 \\ -0.0305 & 0 & 0.000404 \\ 0 & 0 & 0 \\ 0 & -0.00187 & 0 \end{bmatrix}$$

Our controller's enhanced smooth trajectory is attributed to its capability in accurately and instantaneously estimating complex disturbances and responding intelligently to them. This high precision is achieved through our DRL agent and two key technologies:

Capsule Network: This model creates a deep and multi-dimensional understanding of wind and wave conditions, modeling the system's state far more accurately than linear models.

Federated Learning: This framework, by leveraging the collective experience of all turbines in a wind farm, renders the estimation model highly robust and generalizable.

Consequently, the combination of this precise disturbance estimation and the guaranteed stability of the SMC yields a more stable and smoother performance compared to the linear PI controller.

The vector $\boldsymbol{X}^T = \begin{bmatrix} x & \theta & \dot{x} & \dot{\theta} & \omega_r & \dot{\omega}_r & \omega_g & \dot{\omega}_g \end{bmatrix}^T$ represents the dynamic states of FWT, which are essential for characterizing its motion and overall behavior. The desired FWT trajectories are generated using inputs from the proposed adaptive controller, and these trajectories serve as a critical benchmark for evaluating the effectiveness of the new control strategy presented in this paper. By employing adaptive laws derived from Lyapunov theory, the controller successfully ensures that the tracking error is minimized, as illustrated in Fig 5. The results demonstrate that the tracking error consistently remains close to zero, thereby confirming both the reliability and the accuracy of the adaptive control method.

Furthermore, a comparative analysis highlights the superior performance of DRL-based approaches over radial basis function neural networks (RBFNNs). In particular, DRL methods exhibit faster convergence rates and improved tracking accuracy, enabling the system to achieve the desired response with minimal fluctuations. This enhanced stability and robustness emphasize the advantages of DRL controllers in managing the complex dynamics of FWT systems, while also underscoring their suitability for practical deployment in real-world scenarios.

In addition, the controller design is rigorously validated by demonstrating that its surfaces asymptotically converge to zero. This behavior is illustrated in Fig 6, where the continuous curves show the temporal evolution of the surfaces. The graphical results clearly confirm that all surfaces converge to zero, thereby validating the robustness of the controller design. In this context, the phrase asymptotically converges to zero is used interchangeably with converges to zero, emphasizing the systematic reduction and quantification of tracking errors. The presented evidence highlights that the controller surfaces were intentionally designed to achieve this property, reinforcing the overall effectiveness of the proposed control strategy.

Furthermore, the comparative results between the RBFNN and the DRL-based control system gives additional support for the advantages of DRL. The results reveal that DRL enables faster tracking with fewer fluctuations, allowing the

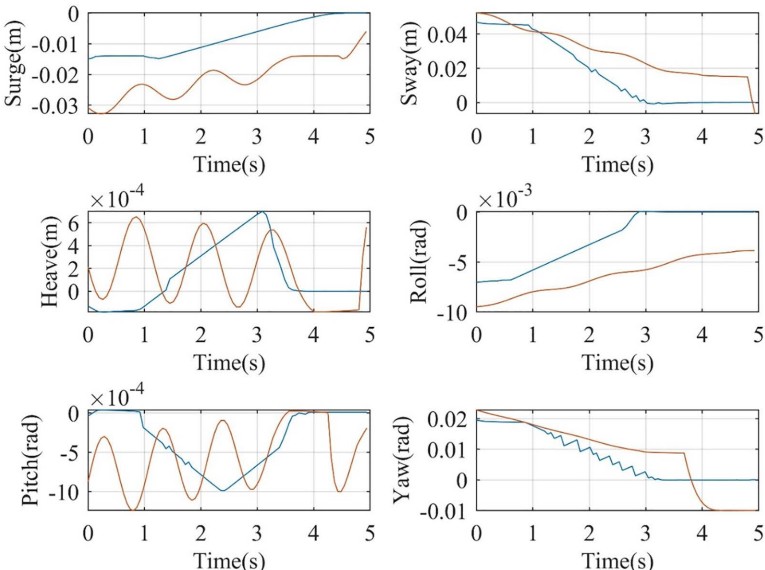

**Fig 5. State tracking errors using RBFNN- and DRL-based controller (red diagrams are RBFNN and blue ones are DRL based).**

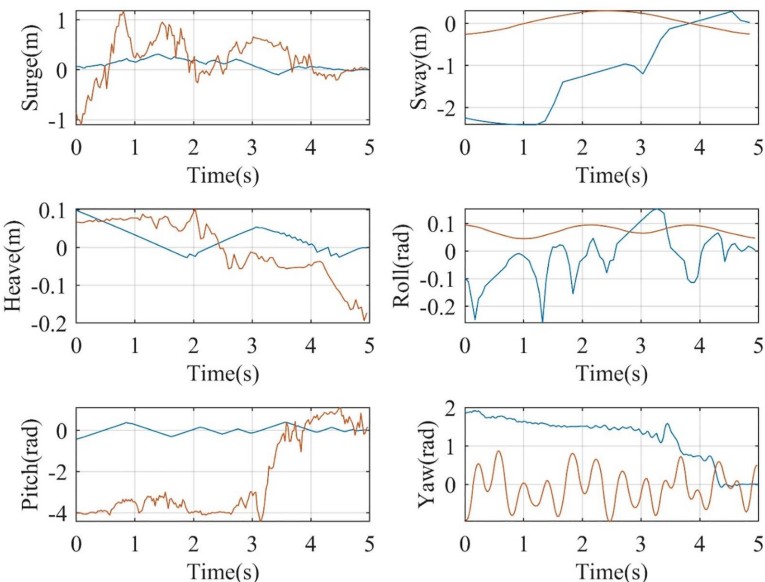

**Fig 6. RBFNN- and DRL-based defined sliding surfaces (red diagrams are RBFNN and blue ones are DRL based).**

system to reach the desired response more efficiently. This reduction in oscillations not only increases tracking accuracy but also extends actuator lifespan by minimizing excessive wear. Taken together, these findings underscore the superiority of DRL in delivering accurate and stable control performance, establishing it as a strong candidate for highly dynamic and complex control systems.

Finally, within the control task framework, three distinct control inputs are required to achieve the results presented in Figs 5 and 6. The corresponding control actions are depicted in Fig 7, which illustrates the nacelle pitch and yaw angles along with the generator torque. Specifically, the pitch angle should exhibit only small fluctuations around zero degrees, the yaw angle should stabilize at an average of around 4 degrees, and the generator torque should oscillate around $1.5 \times 10^4$. Overall, these coordinated control actions yield the performance results highlighted above.

The dynamics of a FWT involve complex interactions between the turbine and the floating platform. To simplify the analysis, a linear model can be employed to capture the fundamental behavior of the system. In this framework, the floating platform can be represented as a spring–damper system, where wave forces and the turbine's mass govern its motion. The turbine itself may be modeled as a rotor system that incorporates both rotor and generator dynamics. Furthermore, the interaction between the platform and the turbine introduces coupling effects, such as the forces exerted by the rotor on the platform and the reactive forces of the platform on the rotor. The motion of the floating platform in response to wave excitation can therefore be expressed by the following equation:

$$\ddot{x}_p + C_p \dot{x}_p + K_p x_p = F_{\text{wave}}(t) - F_{\text{turbine}}(t)$$

where $M_p$ is the mass of the platform, $C_p$ is the damping coefficient, $K_p$ is the spring constant, $x_p$ is the displacement of the platform, $F_{\text{wave}}(t)$ is the external wave force, $F_{\text{turbine}}(t)$ is the force exerted by the turbine on the platform. The dynamics of the turbine rotor can be modeled as:

$$J\ddot{\theta} + C_t \dot{\theta} + K_t \theta = T_{\text{wind}}(t) - T_{\text{platform}}(t)$$

where: $J$ is the moment of inertia of the turbine rotor, $C_t$ is the damping coefficient of the turbine, $K_t$ is the spring constant related to the turbine's stiffness, $\theta$ is the rotational displacement of the turbine rotor, $T_{\text{wind}}(t)$ is the torque from the wind, $T_{\text{platform}}(t)$ is the torque exerted by the platform on the turbine. The coupling effects between the turbine and the platform can be incorporated as:

$$F_{\text{turbine}}(t) = K_{\text{coupling}} x_p$$

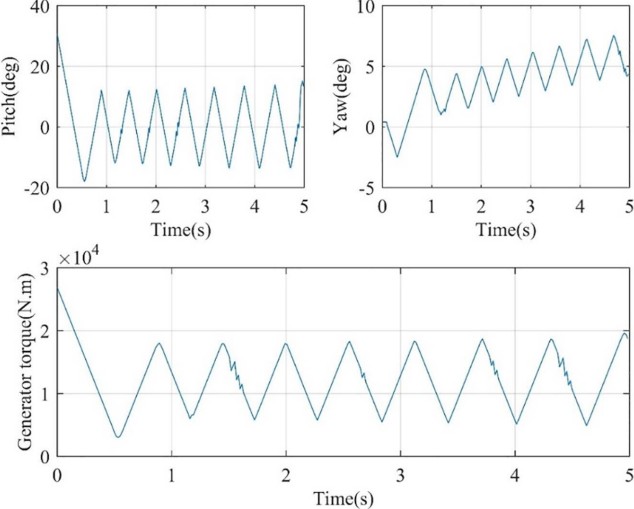

**Fig 7. Pitch and yaw angles of the nacelle and torque of generator as control actions.**

$$T_{\text{platform}}(t) = K_{\text{coupling}}\theta$$

where $K_{\text{coupling}}$ is the coupling stiffness constant.

Assuming small oscillations, we linearize the system around its equilibrium points. The equations of motion become:

$$M_p\ddot{x}_p + C_p\dot{x}_p + K_px_p = F_{\text{wave}}(t) - K_{\text{coupling}}x_p$$

$$J\ddot{\theta} + C_t\dot{\theta} + K_t\theta = T_{\text{wind}}(t) - K_{\text{coupling}}\theta$$

We transform these equations into the frequency domain using either the Laplace or Fourier transform. For simplicity, sinusoidal inputs are assumed, and the steady-state response is derived as:

$$H_p(\omega) = \frac{X_p(\omega)}{F_{\text{wave}}(\omega)} = \frac{1}{M_p\omega^2 + iC_p\omega + K_p - K_{\text{coupling}}}$$

$$H_t(\omega) = \frac{\Theta(\omega)}{T_{\text{wind}}(\omega)} = \frac{1}{J\omega^2 + iC_t\omega + K_t - K_{\text{coupling}}}$$

where $H_p(\omega)$ and $H_t(\omega)$ are the transfer functions for the platform and turbine, respectively, $\omega$ is the angular frequency of the excitation, $X_p(\omega)$ and $\Theta(\omega)$ are the Fourier transforms of $x_p(t)$ and $\theta(t)$, respectively.

The amplitude response of the system is determined by the magnitude of its transfer function:

$$|H_p(\omega)| = \frac{1}{\sqrt{(M_p\omega^2 - K_p + K_{\text{coupling}})^2 + (C_p\omega)^2}}$$

$$|H_t(\omega)| = \frac{1}{\sqrt{(J\omega^2 - K_t + K_{\text{coupling}})^2 + (C_t\omega)^2}}$$

The peak amplitude arises when the denominator is minimized, which depends on the system's natural frequency and damping ratio. The frequency at which this maximum amplitude occurs is referred to as the resonant frequency. To provide deeper insight into the energy consumption of a FWT and its relationship to the system's nonlinear vibration characteristics, Fig 8 presents the spectral distribution of the dynamic response of the uniform structure across all degrees of freedom, plotted on a logarithmic scale. This figure highlights the frequency characteristics of the system, which are critical for understanding its potential in energy recovery. Furthermore, Table 3 lists the extracted natural frequencies, offering a clear reference for the frequencies considered in the analysis. Together, these additions strengthen the demonstration of the analogy between the spring–mass system and the frequency-dependent energy recovery characteristics of FWT.

This study also evaluates the use of MEMS-based systems, piezoelectric biosensors, and periodic MEMS methods for stabilizing wind turbine structures. Aerodynamic and acoustic measurement systems based on MEMS offer considerable benefits for wind turbines because of their small size, excellent sensitivity, and durability in extreme conditions. These MEMS sensors measure aerodynamic forces by analyzing pressure distributions through the equation $F_{aero} = \frac{1}{2}\rho v^2 C_L A$, assuming the known parameters $\rho$, $v$, $C_L$ and $A$ as the density of air, the speed of the wind, the lift coefficient, and the area of the blade, correspondingly. They also detect acoustic signals, where sound intensity is linked to acoustic pressure

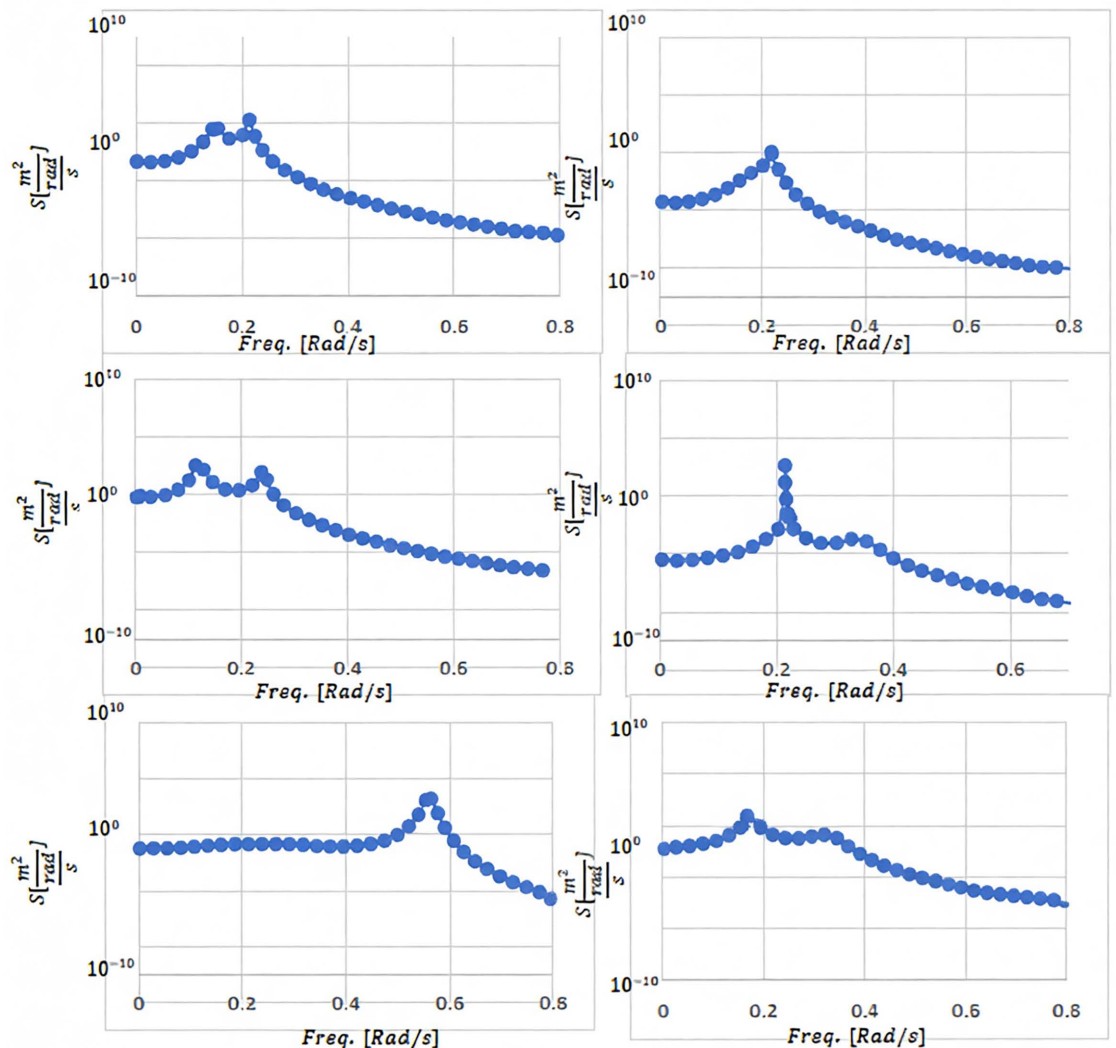

**Fig 8. Spectrum frequency of dynamic response for faultless structure in each degree of freedom (logarithmic scale).**

**Table 3. Extracted natural frequencies.**

| Mode | Natural Frequency [rad/s] |
| --- | --- |
| Surge | 0.1423 |
| Sway | 0.1384 |
| Heave | 0.5607 |
| Roll | 0.222 |
| Pitch | 0.2128 |
| yaw | 0.1754 |

by $I = \frac{P^2}{\rho_0 c}$, where $\rho_0$, $c$ stand for the density of air and the velocity of sound, correspondingly. Data from MEMS accelerometers and microphones are utilized for monitoring vibrations and optimizing control systems, thus improving turbine performance and stability. Piezoelectric biosensors serve to track the structural stability and facilitate energy collection

in floating wind turbines. These sensors transform mechanical strain into electrical signals, with the voltage $V$ expressed as $V = d_p \cdot F$ with $d_p$ denoting the piezoelectric strain coefficient and $F$ the force applied. Furthermore, they are capable of collecting energy, which is determined as $E = \frac{1}{2}CV^2$, assuming $C$ and $V$ as capacitance and voltage, respectively. In FWTs, piezoelectric sensors are crucial for identification of modifications in structural loads and vibrations, and the harvested energy can be derived from small turbine systems, improving efficiency and minimizing maintenance requirements. Grasping periodic solutions in MEMS is crucial for predicting and improving their performance in FWTs, especially in dynamic scenarios like varying wind speeds. Periodic behavior in MEMS can be examined using techniques like harmonic balance and Floquet theory. In a simple harmonic oscillator model, the equation that describes it is a differential equation:

$$m\ddot{x} + c\dot{x} + kx = F_0\cos(\omega t),$$

where $m$ is the mass, $c$ is the damping coefficient, $k$ is the spring constant, $F_0$ is the amplitude of the force, and $\omega$ is the excitation frequency. The steady-state solution is:

$$x(t) = \frac{F_0}{\sqrt{(k - m\omega^2)^2 + (c\omega)^2}}\cos(\omega t - \varphi),$$

with $\varphi$ being the phase shift computed from:

$$\tan(\varphi) = \frac{c\omega}{k - m\omega^2}$$

Analysis of these solutions helps to assess the stability of MEMS devices under regular loads and guides the optimization of design parameters to avoid resonance and to ensure a reliable operation under varying conditions. This analysis is essential for the design of MEMS that can cope with the dynamic loads encountered in the applications of wind turbines.

The resonance peaks of the linearized model over the system's natural frequencies in Table 3 reveal identification of critical structural vulnerabilities corresponding to wind and wave disturbances. The proposed power optimization controller through combining SMC and DRL mitigates these resonant oscillations and hence prevents energy stuck in the system's natural modes. Unlike conventional linear controllers, which are limited to narrow operating ranges, our nonlinear controller ensures stability and robustness across diverse operational conditions through two synergistic mechanisms:

1. SMC Component: Utilizes a well-defined sliding surface to asymptotically drive the system toward equilibrium while maintaining robustness against parametric uncertainties.

2. DRL Component: Dynamically adapts SMC parameters in real time to environmental variations (e.g., fluctuating wind speed and wave height), ensuring optimal oscillation damping at critical frequencies.

To validate the controller's efficacy, we injected a frequency-sweep chirp signal into the closed-loop system and analyzed its frequency response. Key findings from the Bode diagram (Fig 9) include:

• The open-loop system exhibits pronounced resonance peaks at natural frequencies (e.g., Pitch and Surge), indicating high susceptibility to wave-induced disturbances.

• A conventional PID controller normally attenuates these peaks, however our SMC-DRL controller suppresses resonance amplitudes by more attenuation about 20% in across all state variables, Surge, Sway, Heave, Roll, Pitch, Yaw, as depicted in the Resonance Peak Comparison bar graph at bottom-right of Fig 9.

Furthermore, the time-domain response to the chirp signal (bottom-left) demonstrates the controller's adaptability to time-varying disturbances. While the PID controller yields significant oscillations, the SMC-DRL system maintains minimal amplitude deviation, underscoring its superiority in:

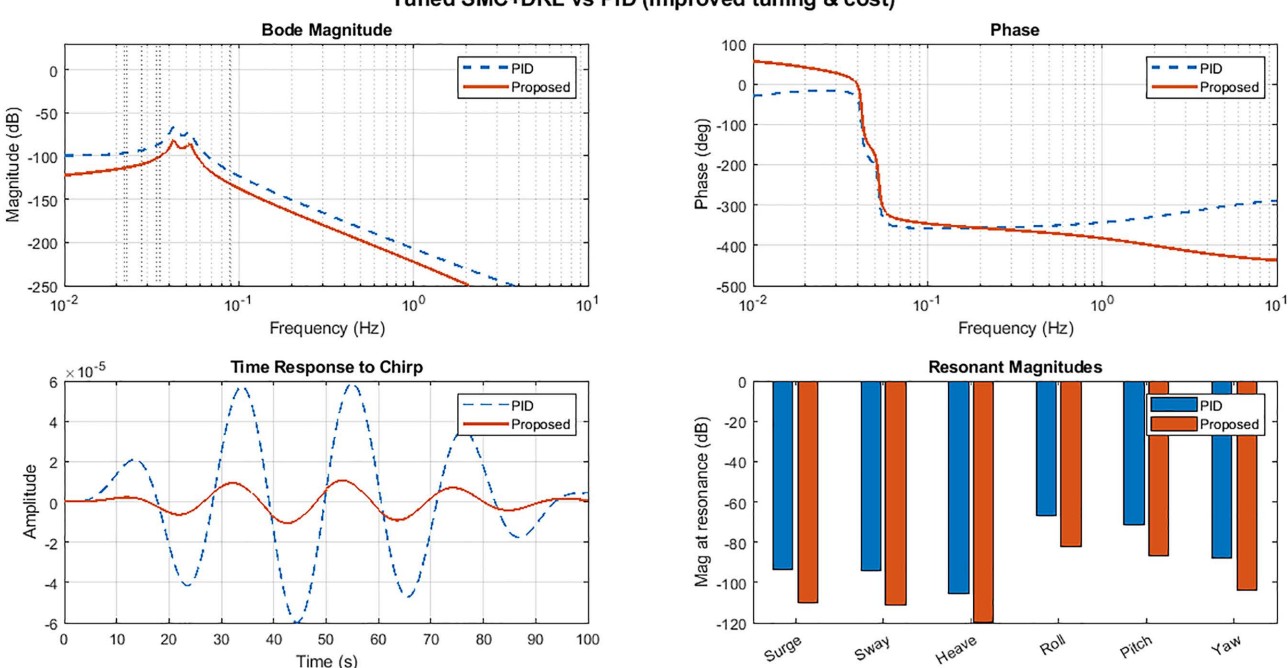

**Fig 9. Performance analysis of the proposed SMC-DRL compared to the PID controller and open-loop mode.**

- Stability margin enhancement (reduced resonance energy transfer),

- Disturbance rejection (critical for power generation optimization under dynamic conditions).

These results conclusively establish the controller's ability to decouple disturbance energy from structural modes, achieving robust performance in realistic marine environments.

## Conclusions

In this study, a novel intelligent control framework was presented to stabilize and optimize the performance of FWT under uncertain environmental situations. The main goal is to overcome the challenges arising from nonlinear system dynamics and unpredictable wind and wave forces to maximize the efficiency and sustainability of clean energy exploitation. Our innovative framework combines the inherent stability of ASMC with the learning and optimal decision-making capabilities of DRL. To increase the accuracy of disturbance detection, a capsule network was used instead of traditional neural networks, which is capable of figuring out spatial and hierarchical relationships between sensor data. Furthermore, by employing federated learning, it is possible to train a robust model in a distributed manner on multiple turbines, which increases the overall accuracy of the system while preserving data privacy. Lyapunov stability analysis mathematically proved that the proposed controller is globally stable and the system tracking error converges to zero asymptotically, which ensures the reliability of the controller under various conditions. Simulation results also clearly agreed with the absolute superiority of this method compared to an advanced neural network-based controller (RBFNN); Our controller achieved faster convergence with less oscillations, which means reduced mechanical wear and increased turbine service life.

These achievements are an important step towards making FWTs a more reliable and efficient energy source. This advanced approach not only improves the stability and performance of turbines, but also has the potential to revolutionize

the management of large-scale renewable energy systems. Despite the promising results, this research can be expanded in the future. It is proposed to evaluate the performance of this controller under extreme sea conditions and also considering time-delays in the control signals. Overall, this paper presents a comprehensive and intelligent solution that can significantly transform the stability and efficiency of the next generation of FWTs, paving the way for a wider exploitation of wind energy in deep waters.

## Supporting information

**S1 File. Uploaded file in Zenodo repository.**
(XLSX)

## Author contributions

**Conceptualization:** Hadi Mohammadian KhalafAnsar, Jafar Keighobadi, Mohsen Shahhosseini.

**Data curation:** Hadi Mohammadian KhalafAnsar, Jafar Keighobadi, Mohsen Shahhosseini.

**Formal analysis:** Hadi Mohammadian KhalafAnsar, Jafar Keighobadi, Mohsen Shahhosseini.

**Investigation:** Hadi Mohammadian KhalafAnsar, Jafar Keighobadi, Mohsen Shahhosseini.

**Methodology:** Hadi Mohammadian KhalafAnsar, Jafar Keighobadi, Mohsen Shahhosseini.

**Resources:** Hadi Mohammadian KhalafAnsar, Jafar Keighobadi, Mohsen Shahhosseini.

**Software:** Hadi Mohammadian KhalafAnsar, Jafar Keighobadi, Mohsen Shahhosseini.

**Supervision:** Jafar Keighobadi.

**Validation:** Hadi Mohammadian KhalafAnsar, Jafar Keighobadi, Mohsen Shahhosseini.

**Visualization:** Hadi Mohammadian KhalafAnsar, Jafar Keighobadi, Mohsen Shahhosseini.

**Writing – original draft:** Hadi Mohammadian KhalafAnsar, Jafar Keighobadi, Mohsen Shahhosseini.

**Writing – review & editing:** Hadi Mohammadian KhalafAnsar, Jafar Keighobadi, Mohsen Shahhosseini.

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
