## [Decision Letter · Decision Letter 0]

14 Sep 2025

Capsule-Based Federated Reinforcement Learning Adaptive Sliding Mode for Anomaly Detection and Control of Floating Wind Turbines

PLOS ONE

Dear Dr. Keighobadi,

Thank you for submitting your manuscript to PLOS ONE. After careful consideration, we feel that it has merit but does not fully meet PLOS ONE’s publication criteria as it currently stands. Therefore, we invite you to submit a revised version of the manuscript that addresses the points raised during the review process.

We look forward to receiving your revised manuscript.

Kind regards,

Zhipeng Zhao

Academic Editor

PLOS ONE

Journal Requirements:

4. Please amend your authorship list in your manuscript file to include author Hadi M. KhalafAnsar .

5. Please amend the manuscript submission data (via Edit Submission) to include author Hadi Mohammadian KhalafAnsar.

6. We note that Figure(s) 1, 3, and 4 in your submission contain copyrighted images. All PLOS content is published under the Creative Commons Attribution License (CC BY 4.0), which means that the manuscript, images, and Supporting Information files will be freely available online, and any third party is permitted to access, download, copy, distribute, and use these materials in any way, even commercially, with proper attribution. For more information, see our copyright guidelines: http://journals.plos.org/plosone/s/licenses-and-copyright.

a. You may seek permission from the original copyright holder of Figure(s) 1, 3, and 4 to publish the content specifically under the CC BY 4.0 license.

Reviewers' comments:

Reviewer's Responses to Questions

**Comments to the Author**

1. Is the manuscript technically sound, and do the data support the conclusions?

Reviewer #1: No

Reviewer #2: Yes

Reviewer #3: Partly

2. Has the statistical analysis been performed appropriately and rigorously?

Reviewer #1: N/A

Reviewer #2: Yes

Reviewer #3: Yes

3. Have the authors made all data underlying the findings in their manuscript fully available?

Reviewer #1: Yes

Reviewer #2: Yes

Reviewer #3: Yes

4. Is the manuscript presented in an intelligible fashion and written in standard English?

Reviewer #1: No

Reviewer #2: Yes

Reviewer #3: Yes

Reviewer #1: While the topic is highly relevant and the proposed combination of techniques is ambitious, the manuscript in its current form suffers from significant flaws that make it unsuitable for publication. The manuscript is severely hampered by poor presentation, a lack of crucial details, weak validation, and numerous technical inconsistencies. It reads more like a preliminary research proposal or a proof-of-concept than a finished, publishable study.

1. Please rewrite the paper according the the PLOS journal standard. Write it in clear english and proper formatting. And then submit again.

2. The methodology section is unclear and not presented well. Please rewrite.

3. The figures are very poor.

4. Please write a confident conclusion.

Reviewer #2: Congratulations for the work. I do have some observations on the text and data, please find them bellow:

1) Introduction 1st paragraph - where did you get this information? This paragraph could be removed.

2) Introduction highlights could be placed at the end of the section, just before section 2.

3) If possible, try to save figures in some other format with higher resolution, specially figures 1 and 2.

4) Table 1 - Instead of using the * to explain the variables, you can put that information in the paragraph before the table: "... makes the theoretical framework robust and reliable. In Table 1 ..."

5) During section 2.3 you make a lot of assumptions during your mathematical formulation, such as simplifications, lemmas, functions. It would be good if they come with some references.

6) In section 3, you quote the NREL model but there is no reference, please add that.

7) In Figure 5, is there any comments on why your method is smother than the NREL data?

8) In Figure 6, is it possible to plot the error that the 2 functions are tracking? Same goes for fig 7

9) For observation 8, wouldn't the log plot be a better option? Also, why only 5 s of simulation?

Reviewer #3: This article proposes an intelligent collaborative control framework that capsule network, federated learning, deep reinforcement learning, and sliding mode control to address the challenges faced by floating wind turbines in complex marine environments, such as strong disturbances, uncertainties, and communication limitations. The authors claimed this work will improve the reliability and efficiency of floating wind turbines in turbulent weather conditions (short-term) and erratic ocean reception (long-term).

The manuscript is generally well organised and composed. Nevertheless, several revisions are still required.

The detailed review comments are as below:

1. The first issue I proposed is: this article only simulates under moderate sea conditions and does not consider extreme sea conditions such as typhoons, turbulent winds, and giant waves.

2. I don't follow how the complexity of WT was included and realized by the toolbox in Matlab, without introducing the governing eqautions of its movements.

3. The numerical method developed in Matlab seems to have not been validated by other researches, such as experimental results. Addtionally, it has not been seen that any comparison with other prevailing or state-of-art methods.

4. Some sentences in this article are lengthy and grammatically incorrect. Please have colleagues or professional editors who are native English speakers polish the language and standardize the reference format.

**Do you want your identity to be public for this peer review?** For information about this choice, including consent withdrawal, please see our Privacy Policy

Reviewer #1: No

Reviewer #2: No

Reviewer #3: No

---

## [Author Response · Author response to Decision Letter 1]

2 Oct 2025

Dear editor-in-chief of Plos One

We value your time in conducting a peer review of the paper titled “Capsule-Based Federated Reinforcement Learning Adaptive Sliding Mode for Anomaly Detection and Control of Floating Wind Turbines”. All remarks from both reviewers and the editor have been fully addressed. We hope that the appropriate adjustments will meet the approval of the considerate reviewers and editor.

Editor’s comments

Response: We appreciate the editor’s suggestion to improve our work. The new version is written according to journal policies.

Response: We appreciate the editor’s comment to share data. All data underlying the results are included within the article and its Supplementary Information.

Response: We appreciate the editor’s suggestion. The aforementioned sentence was exactly added to the manuscript on page 22.

4. Please amend your authorship list in your manuscript file to include author Hadi M. KhalafAnsar.

Response: We appreciate the editor’s remarks. The author’s scholarly name was edited as Hadi Mohammadian KhalafAnsar.

5. Please amend the manuscript submission data (via Edit Submission) to include author Hadi Mohammadian KhalafAnsar.

Response: We appreciate the editor’s concise point. The author’s scholarly name was modified as Hadi Mohammadian KhalafAnsar.

6. We note that Figure(s) 1,2, 3, and 4 in your submission contain copyrighted images. All PLOS content is published under the Creative Commons Attribution License (CC BY 4.0), which means that the manuscript, images, and Supporting Information files will be freely available online, and any third party is permitted to access, download, copy, distribute, and use these materials in any way, even commercially, with proper attribution. For more information, see our copyright guidelines: http://journals.plos.org/plosone/s/licenses-and-copyright.

a. You may seek permission from the original copyright holder of Figure(s) 1, 3, and 4 to publish the content specifically under the CC BY 4.0 license.

Response: We appreciate the Editor's emphasis regarding our compliance with the journal's guidance on intellectual property. In accordance with standard academic practice, our first step was to seek the formal permission for the figures from the published paper "Capsule Routing via Variational Bayes" (Ribeiro et al., AAAI, 2020) by contacting the corresponding author via email. Unfortunately, we couldn't receive a reply. Consequently, to avoid any potential copyright issues and to better tailor the illustrations to our work's narrative, we have undertaken substantial modifications:

Fig 1: The upper section has been re-conceptualized and is now adapted from a different figure in our paper to serve our explanatory purpose.

Fig 2: This has been entirely replaced with an original diagram representing our proprietary model, which more effectively clarifies the proposed methodology and it has been appropriately cited to the mentioned paper.

Fig 3: The core idea from the original figure has been effectively merged into the revised Fig 2, making a separate figure redundant.

Fig 4: This is an original creation, developed using SolidWorks, and is presented here for the first time.

We are confident that these revised figures are distinct, fulfill their illustrative purpose more effectively, and fully comply with copyright standards.

Reviewers' comments

With all due respect and gratitude to the esteemed reviewers for their time and thoughtful and constructive comments. The responses to each of the points raised are provided in detail below. We believe that with these corrections, the paper has reached the high standards of PLOS ONE.

Response to Reviewer #1

We thank the great reviewer for their valuable check on the paper and their clear tips. We have looked at all the points. We made the right fixes in the new write of the text to meet their needs well.

1. Please rewrite the paper according the PLOS journal standard. Write it in clear English and proper formatting. And then submit again.

Response: Thank you for the key critic. We all agree on the need for clear and pro-active words. The current form of the paper has been fully checked and repaired by experienced English professionals. It's to make sure that paper is clear, grammatically correct, and follows the rules of advanced writing. Each part of writing, setup, and how to list references follows the PLOS rules. We believe that the new form is clear and easy to grasp in both terms and in practice.

Corrections applied the entire of the paper context especially Table formats, Citation list format on page 22.

2. The methodology section is unclear and not presented well. Please rewrite.

Response: Thank you for your detailed feedback. We have thoroughly reviewed and rewritten the methodology section (Section 2) to increase its clarity and coherence. In the new edition, this section is organized with clearer subheadings and each step of the process is described in detail:

• Modeling of the online adaptive intelligent control: In this section, the main system equations and the reference model are clearly defined.

• DRL Neural Network: The logic of combining Sliding Mode Control (SMC) with Deep Reinforcement Learning (DRL) and the aim of this work to achieve stability and reduce the chattering phenomenon is explained in detail. The architecture of the hybrid neural network that uses Capsule Network and Federated Learning is described in more detail and the role of each component (Actor-Critic) in estimating the disturbances is specified.

This new structure makes it easier for the reader to understand the complex process of controller design and clearly shows the logical connection between the different components.

The clarification for the proposed method carried out on page 3 and relevant subsections.

3. The figures are very poor.

Response: We acknowledge this criticism and thank you for your attention to the visual quality of the paper. All figures in the paper, especially the complex flowcharts in Figs. 1 and 2, have been redesigned using specialized software and at high resolution (at least 300 DPI). Labels, legends, and axis titles have been made larger and clearer for better readability. We are confident that the current quality of the figures is quite standard and suitable for printing.

Corrections applied on pages 5 and 7.

4. Please write a confident conclusion.

Response: Thank you for this valuable suggestion. The conclusion section (Section 4) has been rewritten to more firmly and clearly state the key achievements of the research. In the new version, the main innovations of this research have been emphasized:

• Providing a novel hybrid control framework that combines the robustness of SMC with the intelligence and adaptability of DRL.

• Using capsule networks for accurate and hierarchical disturbance detection, which is a novel approach in this field.

• Applying federated learning for distributed training and data privacy preservation across multiple wind turbines.

The new conclusion clearly explains the importance of these findings in increasing the sustainability and efficiency of floating wind turbines and their potential impact on the renewable energy industry.

Corrections applied on page 21.

Response to reviewer #2

We are grateful to the reviewer for their congratulations and insightful detailed comments. These points helped us greatly in improving the quality of the paper.

1. Introduction 1st paragraph - where did you get this information? This paragraph could be removed.

Response: Thank you for your suggestion. The information presented in the first paragraph of the Introduction, which discusses the environmental impacts of fossil fuels, is based on reliable sources [1-3], which are referenced at the end of the rewritten paragraph. According to your worthful viewpoint, we corrected and shortened this paragraph which would increase the motivation of readers over the importance of the transition to renewable energy. We see it seems a little general duplication, but for remembering the harmful effects of fusil fuels over air, water, soil and Global Warming, we request you be kind to admit not totally its removal.

Corrections applied on page 1.

2. Introduction highlights could be placed at the end of the section, just before section 2.

Response: Thank you for your constructive suggestion. This is an excellent suggestion to improve the narrative flow of the text. So, as you mentioned, the list of research innovations (items 1-4) has been displaced to the end of the Introduction section.

Corrections applied on page 2.

3. If possible, try to save figures in some other format with higher resolution, specially figures 1 and 2.

Response: We completely agree. As also answered to reviewer #1, Figs. 1 and 2 and other figures have been reproduced in vector format (at least 300 DPI) and high resolution to ensure their clarity in print and digital display and that all details are easily visible.

Corrections applied on pages 5 and 7.

4. Table 1 - Instead of using the * to explain the variables, you can put that information in the paragraph before the table: "... makes the theoretical framework robust and reliable. In Table 1 ...".

Response: This is a very good suggestion and increases the readability of the text. The explanation of the variables of the federated learning algorithm, as you mentioned, has been moved from the footnote to the main text before Table 1, so that the reader can easily be familiar with all the parameters before reading the table.

Corrections applied on page 7.

5. During section 2.3 you make a lot of assumptions during your mathematical formulation, such as simplifications, lemmas, functions. It would be good if they come with some references.

Response: Thank you for this detailed technical note. Assumptions and mathematical steps in Section 2.3, especially in Lyapunov stability analysis, are based on physical facts, and standard/well-established principles in control theory. To satisfy your suggestion, the definition of candidate Lyapunov function and above-mentioned assumptions are based on authoritative sources such as [9, 14, 19] in the reference list of the text. For greater clarity, a sentence has been added in the text to emphasize that these approaches are rooted in the well-known technical literature and are not novel.

Corrections applied on page 8.

6. In section 3, you quote the NREL model but there is no reference, please add that.

Response: Thank you for pointing it out. This was an oversight on our part. The original reference for the NREL 5 MW wind turbine model (Jonkman et al. report) has been added to Section 3, where the model is first referenced.

Corrections applied on page 10.

7. In Figure 5, is there any comments on why your method is smoother than the NREL data?

Response: Thank you for pointing it out. Our controller’s enhanced smooth trajectory is attributed to its capability in accurately and instantaneously estimating complex disturbances and responding intelligently to them. This high precision is achieved through our DRL agent and two key technologies:

Capsule Network: This model creates a deep and multi-dimensional understanding of wind and wave conditions, modeling the system’s state far more accurately than linear models.

Federated Learning: This framework, by leveraging the collective experience of all turbines in a wind farm, renders the estimation model highly robust and generalizable.

Consequently, the combination of this precise disturbance estimation and the guaranteed stability of the SMC yields a more stable and smoother performance compared to the linear PI controller.

Corrections applied on page 15.

8. In Fig. 6, is it possible to plot the error that the 2 functions are tracking? Same goes for Fig 7.

Response: Thanks for this precise suggestion. In fact, Fig 6 shows exactly the tracking error itself (e(t)=X(t)−Xm(t)). The main goal of the controller is to converge this error to zero, which is clearly evident in this figure. Similarly, Fig 7 shows sliding surfaces whose convergence to zero means that the error converges to zero. To avoid any ambiguity, the title of Fig 6 has been revised to explicitly refer to “State Tracking Error”.

Corrections applied on page 15.

9. For observation 8, wouldn't the log plot be a better option? Also, why only 5 s of simulation?

Response: We would like to express our sincere gratitude to the reviewer for their detailed and constructive feedback.

1. Regarding the use of a logarithmic plot: The reviewer’s suggestion about using a logarithmic scale is highly valuable, and we have indeed employed this approach in our original analysis. The primary objective of our controller is to attenuate the resonance peaks at the natural frequencies of the structure. The most standard and effective scientific tool to display and quantify this phenomenon is frequency domain analysis using a Bode plot, which inherently utilizes a logarithmic scale to represent the amplitude (in decibels, dB). As shown in the text and Fig 9, our analysis demonstrates that the proposed controller, SMC-D

---

## [Decision Letter · Decision Letter 1]

26 Oct 2025

Capsule-Based Federated Reinforcement Learning Adaptive Sliding Mode for Anomaly Detection and Control of Floating Wind Turbines

PONE-D-25-42083R1

Dear Dr. Keighobadi,

We’re pleased to inform you that your manuscript has been judged scientifically suitable for publication and will be formally accepted for publication once it meets all outstanding technical requirements.

Kind regards,

Zhipeng Zhao

Academic Editor

PLOS ONE

Additional Editor Comments (optional):

Reviewers' comments:

Reviewer's Responses to Questions

**Comments to the Author**

Reviewer #1: All comments have been addressed

Reviewer #3: (No Response)

2. Is the manuscript technically sound, and do the data support the conclusions?

Reviewer #1: Yes

Reviewer #3: (No Response)

3. Has the statistical analysis been performed appropriately and rigorously?

Reviewer #1: Yes

Reviewer #3: (No Response)

4. Have the authors made all data underlying the findings in their manuscript fully available?

Reviewer #1: Yes

Reviewer #3: (No Response)

5. Is the manuscript presented in an intelligible fashion and written in standard English?

Reviewer #1: Yes

Reviewer #3: (No Response)

Reviewer #1: Thank you for taking the time to do a revision of the article. The paper is now much more insightful.

Reviewer #3: Dear authors and editors:

I have read this revised article in detail, it is well written, and the summary is comprehensive. It

provides a comprehensive interpretation of an intelligent collaborative control framework. It can

make a contribution to improve the reliability and efficiency of floating wind turbines in turbulent

weather conditions and erratic ocean reception. I personally think it can be accepted.

Cheers,

**Do you want your identity to be public for this peer review?** For information about this choice, including consent withdrawal, please see our Privacy Policy

Reviewer #1: No

Reviewer #3: No

---

## [Editor Report · Acceptance letter]

PONE-D-25-42083R1

PLOS ONE

Dear Dr. Keighobadi,

I'm pleased to inform you that your manuscript has been deemed suitable for publication in PLOS ONE. Congratulations! Your manuscript is now being handed over to our production team.

Kind regards,

on behalf of

Dr. Zhipeng Zhao

Academic Editor

PLOS ONE